# Tetrahydrobiopterin as a Trigger for Vitiligo: Phototransformation during UV Irradiation

**DOI:** 10.3390/ijms241713586

**Published:** 2023-09-01

**Authors:** Taisiya A. Telegina, Yuliya L. Vechtomova, Vera A. Borzova, Andrey A. Buglak

**Affiliations:** 1Research Center of Biotechnology of the Russian Academy of Sciences, Bach Institute of Biochemistry, 119071 Moscow, Russia; telegina@inbi.ras.ru (T.A.T.); vechtomova@inbi.ras.ru (Y.L.V.); vera.a.borzova@gmail.com (V.A.B.); 2Faculty of Physics, Saint Petersburg State University, 199034 Saint Petersburg, Russia

**Keywords:** vitiligo, tetrahydrobiopterin, oxidative stress, H_2_O_2_, UVB vitiligo phototherapy, photooxidation, Gibbs free energy, dihydropterin dimers

## Abstract

Vitiligo is a type of hypomelanosis. Tetrahydrobiopterin (H_4_Bip), the coenzyme of the initial stage of melanogenesis, appears to be a trigger for vitiligo. H_4_Bip is present in vitiligo in 3–5-fold excess and causes oxidative stress by triggering an autocatalytic cycle of excess hydrogen peroxide synthesis. Using quantum-chemical calculations, we have evaluated the possibility of H_4_Bip reactions occurring in the dark and under ultraviolet (UV) irradiation, including the formation of dihydropterin dimers. In order to simulate the oxidative stress, oxidative modification of human serum albumin (HSA) has been carried out in the presence of excessive H_4_Bip using the fluorescence method. The fraction of oxidized protein (FOP) has been calculated. It has been established that there is a strong oxidative modification of amino acids chromophores (tryptophan and tyrosine) in the protein (FOP 0.64). Under UV irradiation of the system (HSA + H_4_Bip), FOP is reduced to 0.39. Apparently, a part of H_4_Bip transforms into dihydropterin dimers and does not participate in the oxidative modification of the protein. The data on oxidative modification of HSA are consistent with dynamic light scattering: H_4_Bip promotes HSA aggregation with the formation of particles with a hydrodynamic radius *R*_h_ ≥ 2000 nm, which can become immunogenic.

## 1. Introduction

Vitiligo is a chronic dermatological disease characterized by the formation of depigmented patches on the skin due to impaired melanin pigment biosynthesis [1,2,3]. The incidence of vitiligo varies by country, from 0.1% to 4.0% [4,5,6,7]. An increase in the incidence of this disease is observed, which determines the relevance of this study.

The occurrence of disorders of melanogenesis in melanocytes, apparently, is associated with the functioning of tetrahydrobiopterin (H_4_Bip), which is a coenzyme of phenylalanine hydroxylase (phenylalanine-4-monooxygenase, EC 1.14.16.1) [1,2,8,9,10,11,12,13,14]. In melanocytes, tyrosine is formed during the hydroxylation of phenylalanine with the participation of the H_4_Bip coenzyme; then, tyrosine is converted to dihydroxyphenylalanine (DOPA) by tyrosinase (EC 1.14.18.1), and then dopachrome is formed on the way to melanin (Figure 1). In vitiligo, melanocytes have a 3–5-fold excess of H_4_Bip, which inhibits tyrosinase, a key enzyme in melanin synthesis [2,15,16,17,18]. Figure 1 shows the cycle of enzymatic regeneration of H_4_Bip during the process of melanogenesis, which can be disturbed under conditions of oxidative stress in vitiligo [15,19,20,21,22,23]. Epidermal biosynthesis of H_4_Bip from GTP is controlled by hormones and cytokines and may be increased by interferon gamma in vitiligo [11,15]. Being a reduced compound, H_4_Bip is easily oxidized by atmospheric oxygen (the so-called autoxidation) both in vitro and in vivo [24,25].

The formation of the H_4_Bip oxidized derivatives is accompanied by the formation of hydrogen peroxide (H_2_O_2_) (Figure 2). Rancy et al. (2020) [26] reviewed 5147 publications in which H_2_O_2_ was used to trigger oxidative stress. It was observed that in half of the publications, the H_2_O_2_ concentration in the range of 100–500 µmol was used to trigger cellular oxidative stress. In vitiligo, H_2_O_2_ was identified in the skin of patients in millimolar concentrations [15,18,19,23,27], which means that there is strong oxidative stress.

We have shown that ultraviolet (UV) irradiation enhances the oxidation of H_4_Bip, both due to the excitation of H_4_Bip itself (λ_max_ 298 nm) and the excitation of biopterin (Bip, λ_max_ 346 nm), the H_4_Bip oxidation product. Irradiation in the absorption region of Bip triggers the process of photosensitized oxidation of H_4_Bip. Such photooxidation of H_4_Bip can lead to the accumulation of an additional amount of H_2_O_2_ and thus prolong vitiligo [29,30,31].

UVB (280–320 nm) phototherapy using UV at 308 and 311 nm is the most successful phototherapy for vitiligo [18,32], but the mechanism of the therapeutic effect has long remained unclear. We have previously proposed a hypothesis and presented arguments in its favor, according to which the main target of UVB radiation is H_4_Bip [33]. The formation of dimers of the azocyclobutane type during UVB photooxidation of H_4_Bip has been shown for the first time. The photoformation of dihydropterin dimers ((H_2_Ptr)_2_) from H_4_Bip has been proven, and it has been shown that the action spectrum of UV radiation lies in the region of 300–325 nm [28].

The study of the H_4_Bip autoxidation process and the literature data on its excessive synthesis in vitiligo allowed us to conclude that the pathology of vitiligo is based on the formation of an autocatalytic cycle of excessive H_2_O_2_ synthesis (Figure 2). Under conditions of oxidative stress, H_2_O_2_ can trigger an autocatalytic cycle through cytokines, in particular γ-interferon. γ-interferon activates the interferon-inducible enzyme, GTP-cyclohydrolase I (EC 3.5.4.16), which synthesizes an excess of H_4_Bip. UVB phototherapy of vitiligo will break the autocatalytic cycle of excess H_2_O_2_ synthesis due to the removal of excess H_4_Bip in the form of (H_2_Ptr)_2_ dimers. Therefore, such UVB therapy will help to restore the process of melanogenesis.

Since autoxidation and photooxidation of H_4_Bip are radical processes, in this paper, using the methods of quantum chemistry, a theoretical study of the possibility of the occurrence of reactions involved in the processes of impaired melanogenesis and UV therapy of vitiligo was carried out. In addition to studying the radical processes of autoxidation and photooxidation of H_4_Bip in buffer solutions, reactions were carried out in the presence of human serum albumin (HSA) to analyze the effect of radicals on proteins, including melanogenesis enzymes. HSA was used as a model for the H_4_Bip protein environment under oxidative stress conditions in cells. Further development in this direction is necessary to improve the methods for diagnosing and treating vitiligo.

## 2. Results

### 2.1. Quantum Chemical Calculations: Conformational Analysis, Geometry Optimization, and Hessian Calculation

#### 2.1.1. H_4_Bip Autoxidation

H_4_Bip is prone to autoxidation in the presence of molecular oxygen. The autoxidation starts with an electron transfer from H_4_Bip to O_2_, which is accompanied by tetrahydrobiopterin radical cation (H_4_Bip^•+^) and superoxide radical anion (O_2_^•−^) production [34]:H_4_Bip + O_2_ → H_4_Bip^•+^ + O_2_^•−^(1)

Autoxidation can also be initiated with the production of trihydrobiopterin radical (H_3_Bip^•^) and hydroperoxyl radical (HOO^•^):H_4_Bip + O_2_ → H_3_Bip^•^ + HOO^•^
(2)

O_2_^•−^ and HOO^•^, which are produced in Reactions (1) and (2), do take part in the oxidation of H_4_Bip. As was previously established, the oxidation of H_4_Bip in the presence of O_2_ in aqueous solutions has a radical chain character [34]. The main products of H_4_Bip autoxidation are well-known. However, the nature and the role of short-lived intermediate free radical species is debatable. The same is true for the photooxidation of H_4_Bip.

The photooxidation of H_4_Bip can occur with the participation of the excited pterin triplets (^3^Ptr*) [29,35]. Oxidized pterins can attract an electron from H_4_Bip (Reaction (3)). As a result, the Ptr^•−^ radical anion and the H_4_Bip^•+^ radical cation are produced. This reaction relates to type-I sensitization mechanism:^3^Ptr* + H_4_Bip → Ptr^•−^ + H_4_Bip^•+^
(3)

Pterin triplets can oxidize not only H_4_Bip, but also other electron donors. Electron transfer from an enzyme to a pterin triplet can inactivate the enzyme, in particular tyrosinase [36], which becomes a significant problem in vitiligo disease.

We obtained the vertical ionization potential (VIP) of H_4_Bip to test the legitimacy of our theoretical approach. According to the MP2/6-31G(d,p)+ calculation, VIP is equal to 4.62 eV. The value of 4.82 eV was estimated using the dielectric constant of solvent, epsilon, which was equal to 80, in the study by Gogonea et al. [37]. According to the calculations carried out using the M06-2X/6-31G(d,p)+ method, VIP was equal to 4.73 eV [38]. Therefore, our MP2/6-31G(d,p)+ calculations are in agreement with previous studies. VIP is an important characteristic which determines various pterin properties, including singlet oxygen generation efficacy [30]. In addition, we determined the vertical electron affinity (VEA) of ^3^O_2_. According to our calculations, it is equal to −3.43 eV, whereas the experimentally determined value is equal to −3.5 eV [39]. In case of the M06-2X calculations, the VEA of molecular oxygen was equal to −3.41 eV [38]. For this reason, our MP2/6-31G(d,p)+ method using the COSMO solvent model is an adequate and legitimate approach.

The experimental study of free radical species and their reactions requires application of techniques such as electron paramagnetic resonance, which is time- and labor-consuming. However, quantum chemistry methods allow for evaluating the feasibility of such reactions theoretically. We started with the feasibility estimation for the reaction between H_4_Bip and molecular oxygen (Reaction (1)). The Gibbs free energy of reactions **(∆_f_Gº)** is equal to 26.5 kcal mol^−1^, according to our calculations. This means that Reaction (1) does not occur spontaneously.

Reaction (2) occurs between the same reagents as Reaction (1) and produces other radicals: H_3_Bip^•^ and HOO^•^. Production of H_3_Bip^•^ and HOO^•^ should not occur as well: **∆_f_Gº** is equal to 30.8 kcal mol^−1^. Nevertheless, H_3_Bip^•^ and HOO^•^ are not the final products. Next, the quinonoid 6,7-dihydrobiopterin (qH_2_Bip) and H_2_O_2_ formation occurs: H_4_Bip + O_2_ → H_2_O_2_ + qH_2_Bip (4)

The Gibbs free energy of Reaction (4) is −4.3 kcal mol^−1^, which means that the reaction between H_4_Bip and molecular oxygen could take place spontaneously. Since 7,8-dihydrobiopterin (H_2_Bip) is 15.3 kcal mol^−1^ more stable than qH_2_Bip, the final pair of products of the reaction between H_4_Bip and molecular oxygen should be H_2_Bip and hydrogen peroxide: H_4_Bip + O_2_ → H_2_O_2_ + H_2_Bip (5)

Reaction (5) possesses a Gibbs free energy equal to −19.5 kcal mol^−1^. Figure 3 shows the outline of the reaction process between H_4_Bip and molecular oxygen. The energetic characteristics of the intermediate products are insignificant, since only the Gibbs free energy of initial reagents and final products is important (−19.5 kcal mol^−1^). For this reason, the autoxidation of H_4_Bip does occur spontaneously. As a whole, there exist a general force for H_4_Bip autoxidation, despite several uphill elementary reactions. Uphill reactions do take place, but possess slow kinetics: in particular, the experimental rate constant of Reaction (1) is equal to 0.6 M^−1^ s^−1^ [34].

There is an alternative to H_2_Bip formation during Reaction (5): 7,8-dihydropterin (H_2_Ptr) and 2-hydroxypropanal can also be formed. However, these products are 5.1 kcal mol^−1^ less stable than H_2_Bip, which is confirmed with the experiments conducted in phosphate buffer pH 7.3 [29].

During H_4_Bip autoxidation, it can be oxidized by reactive oxygen species (ROS). See, for example, Reactions (6) and (7):H_4_Bip + O_2_^•−^ → H_3_Bip^•^ + HOO^−^
(6)
H_4_Bip + HOO^•^ → H_3_Bip^•^ + H_2_O_2_
(7)

The rate constant of Reaction (6) is known to be 3.9 × 10^5^ M^−1^ s^−1^ [34]; therefore, it is extremely fast. The Gibbs free energy of Reaction (6) is 7.3 kcal mol^−1^; therefore, it is unfavorable. On the contrary, a synonymic reaction, Reaction (7), is feasible: **∆_f_Gº** is equal to −18.0 kcal mol^−1^.

The H_4_Bip free radical species also take part in autoxidation. Thus, H_3_Bip^•^ and molecular oxygen participate as educts in Reaction (8):H_3_Bip^•^ + ^3^O_2_ → H_2_Bip + HOO^•^
(8)

This reaction is a rate limiting step in H_4_Bip prolonged oxidation with a reaction rate constant equal to 3.2 × 10^3^ M^−1^ s^−1^ [34]. The reaction is possible since its Gibbs free energy is negative (−1.5 kcal mol^−1^). The production of a short-living complex between H_3_Bip^•^ and O_2_ is also feasible [34]:H_3_Bip^•^ + ^3^O_2_ → H_3_BipOO^•^
(9)

However, Reaction (9) possesses a positive **∆_f_Gº** value (13.5 kcal mol^−1^). It is still feasible, since the overall reaction (Reaction (8)) has a negative **∆_f_Gº**.

The trihydrobiopterin radical, along with O_2_^•−^, can participate in the production of the H_3_BipOO^−^ intermediate complex due to Reaction (10):H_3_Bip^•^ + O_2_^•−^ → H_3_BipOO^−^(10)
which possesses a Gibbs free energy value equal to −15.7 kcal mol^−1^. Then, the short-living intermediate complex can be protonated:H_3_BipOO^−^ + H_3_O^+^ → H_3_BipOOH + H_2_O (11)

Reaction (11) possesses a significantly negative Gibbs free energy (−42.9 kcal mol^−1^).

Reactions (1), (2), (6) and (7) relate to the initiation of the free radical chain process, whereas Reactions (8)–(10) relate to the chain elongation. Reactions of free radical chain termination are also a part of the H_4_Bip autoxidation process. In particular, the reaction between two H_3_Bip^•^ molecules is feasible both experimentally [34] and theoretically:2H_3_Bip^•^ → H_4_Bip + H_2_Bip(12)

The rate constant of Reaction (12) is 9.3 × M^−1^ s^−1^ [34], whereas **∆_f_Gº** is −32.4 kcal mol^−1^.

The free radical chain process can be terminated when its reactive oxygen species interplay with other ROS. Here are a few reactions:2HOO^•^ → H_2_O_2_ + O_2_
(13)
HOO^•^ + O_2_^•−^ → HOO^−^ + O_2_(14)

The Gibbs free energy of Reaction (13) is −48.9 kcal mol^−1^, and therefore, it is feasible. Reaction (14) possesses **∆_f_Gº** equal to −23.6 kcal mol^−1^ and is also favorable.

Thus, we have considered 14 reactions taking place during H_4_Bip oxidation in the presence of O_2_. All reactions are possible from the viewpoint of the MP2/6-31G(d,p)+ calculation, except for Reaction (6) (formation of H_3_Bip^•^ and HOO^−^ when H_4_Bip and O_2_^•−^ are used as educts).

#### 2.1.2. Type-I H_4_Bip Photooxidation

Along with autoxidation under UV irradiation, tetrahydrobiopterin can also be subjected to photooxidation [29,33]. Oxidized pterins are used as photosensitizers in this process. Oxidized pterins are the final products of H_4_Bip oxidation in aqueous solutions. Pterin (Ptr) forms a singlet excited state (^1^Ptr*) under UV radiation; then, production of ^3^Ptr* is possible through the intersystem crossing:Ptr → ^1^Ptr* → ^3^Ptr* (15)

Oxidation of H_4_Bip by the pterin triplets occurs predominantly (87%) through the type-I photosensitization mechanism [29]. It starts with the reaction between H_4_Bip and ^3^Ptr* (Reaction (3)).

The Gibbs free energy of Reaction (3) is negative (−21.4 kcal mol^−1^) and favorable. Thus, the type-I sensitization mechanism begins with the production of H_4_Bip^•+^ and Ptr^•−^ radicals. In air-equilibrated solutions, Reaction (16), which is the electron transfer from Ptr^•−^ to O_2_, can also occur:Ptr^•−^ + O_2_ → Ptr + O_2_^•−^
(16)

This reaction is possible because the Gibbs free energy of such a reaction is equal to −30.4 kcal mol^−1^. O_2_^•−^ can then oxidize H_4_Bip. Direct electron transfer from ^3^Ptr* to O_2_ and the O_2_^•−^ formation, however, are unfavorable since **∆_f_Gº** of this reaction is positive and equals to 18.6 kcal mol^−1^.

The type-I sensitization mechanism is a part of the free radical chain process of both H_4_Bip autoxidation (“dark” reactions) and photooxidation (Table 1).

Thus, we have evaluated the possibility of more than 15 reactions that relate to H_4_Bip oxidation. The conclusions about the feasibility of certain “dark” and light reactions are in line with our previous studies [31,38]. Next, we turn to the reactions of the dihydropterin dimer formation.

#### 2.1.3. Dihydropterin Dimer Formation

In our previous study [33], we found that the azacyclobutane *cis* isomer (Figure 4) is the most favorable product of dihydropterin photodimerization. However, it is not clear which molecules participate in the dimerization: a quinonoid 6,7-H_2_Ptr (qH_2_Ptr) and benzoid form of 7,8-H_2_Ptr, ground state molecules or excited state triplets? To answer these questions, we have performed quantum-chemical calculations for the respective H_2_Ptr molecules.

First, we have evaluated the feasibility of the dimerization of two ground state molecules:H_2_Ptr + H_2_Ptr → (H_2_Ptr)_2_
(17)

The reaction does not occur since its **∆_f_Gº** is positive and equals to 5.9 kcal mol^−1^. However, when one of the H_2_Ptr molecules is the qH_2_Ptr isomer, Reaction (18) is feasible because **∆_f_Gº** equals to −6.6 kcal mol^−1^:H_2_Ptr + qH_2_Ptr → (H_2_Ptr)_2_
(18)

qH_2_Ptr is 12.5 kcal mol^−1^ less stable than H_2_Ptr. **∆_f_Gº** of the reaction equals to −19.2 kcal mol^−1^ when two qH_2_Ptr molecules are used as educts. Though Reaction (18) is feasible according to the calculations, it is not confirmed experimentally [33]: the dimers are not formed during the autoxidation of H_4_Bip. The reason is probably a low lifetime of the quinonoid and its conversion to benzoic dihydropterin.

**∆_f_Gº** is −58.4 kcal mol^−1^ when one of the H_2_Ptr molecules is in the triplet state:H_2_Ptr + ^3^H_2_Ptr* → (H_2_Ptr)_2_
(19)

Therefore, Reaction (19) is favorable. If one of the molecules is a quinonoid, **∆_f_Gº** is even lower (−65.9 kcal mol^−1^):H_2_Ptr + ^3^qH_2_Ptr* → (H_2_Ptr)_2_
(20)
qH_2_Ptr + ^3^H_2_Ptr* → (H_2_Ptr)_2_
(21)

Reaction (21) is even more feasible than Reaction (20): the Gibbs free energy equals to −71.0 kcal mol^−1^. Thus, Reactions (19)–(21) are all favorable. However, in real experimental conditions, the situation is different: when H_2_Bip is used as an educt of photodimerization, the quantum yield is around 5% [40]. However, when the initial reagent is H_4_Bip, the quantum yield of the dimer formation is ca. 96% [33]. In this regard, Reaction (21) looks like the most favorable among Reactions (19)–(21). The energy profiles of all the described reactions are summarized in Figure 5.

The quinonoid dihydropterin formed during H_4_Bip autoxidation is a highly active compound of quinoid nature and, apparently, forms an intermolecular complex with dihydropterin, the so-called quinhydrone. In quinhydrones, a weak donor–acceptor bond is formed between the quinoid and benzenoid structures. Being bound into an intermolecular complex (H_2_Ptr-qH_2_Ptr), the constituent components are not able to react with each other in the ground state, but are able to react in an excited state when exposed to light. The dimerization reaction is apparently facilitated by the mutual orientation of the quinonoid and benzoic forms of dihydropterins in the complex.

### 2.2. Oxidative Modifications of HSA in the Presence of H_4_Bip

#### 2.2.1. H_4_Bip Irradiation in the Presence and Absence of HSA

Figure 6 shows the results of an experiment on irradiating the H_4_Bip solution in the presence of HSA. As can be seen, during irradiation, the absorption increases in the region of 242 nm, which indicates the formation of dihydropterin dimers (with a yield of about 40 ± 1.8%). In the control without irradiation, H_4_Bip is oxidized only to H_2_Bip and dihydropterin, which coincides with the initial stages of H_4_Bip autoxidation, as shown in Figure 2.

It can be seen from the difference absorption spectrum (Figure 6, inset) that upon irradiation, the compound with the absorption maximum at 307 nm decreases, while the absorption increases in the region of 242 nm, and an inflection appears in the region of 276 nm. This may indicate a decrease in the intermediate intermolecular complex (qH_2_Ptr-H_2_Ptr) and an increase in the amount of the resulting dimer (H_2_Ptr)_2_. An intermediate intermolecular complex can be formed during donor–acceptor interactions of the benzoid form of dihydropterins (H_2_Ptr or H_2_Bip) with the quinonoid form of dihydropterins (qH_2_Ptr or qH_2_Bip). Using derivative UV spectrophotometry, it was found that the resulting intermolecular complex has maximum absorption at 307 nm, which is found in the spectrum of the fourth derivative of the H_4_Bip spectrum after keeping the solution in the presence of atmospheric oxygen for several minutes [41].

Figure 7 shows an experiment on the irradiation of the H_4_Bip solution in a Tris buffer pH 7.0 with stepwise irradiation at 308 ± 10 nm in the presence of atmospheric oxygen. As can be seen, the dihydropterin dimer is formed with a 44 ± 2.1% yield, which practically coincides with the yield of dimer formation upon the irradiation of H_4_Bip in the presence of HSA (40 ± 1.8%). This suggests that the presence of a protein has almost no effect on dimer formation.

#### 2.2.2. Study of HSA Oxidative Modification with Fluorescence

The oxidative modification of HSA was assessed based on the decrease in fluorescence in the region of 350 nm (fluorescence of tryptophan and tyrosine) upon excitation at 280 nm. The light with the spectral range of 308 ± 10 nm, used in the treatment of vitiligo [18,42,43], was considered as the source of oxidative stress. Figure 8a shows the HSA fluorescence spectra before and after irradiation at 308 nm for 16 min. As can be seen from Figure 8a, the fluorescence intensity at 350 nm falls down, which indicates that protein oxidation is taking place.

Figure 8b shows the fluorescence spectra (excitation at 280 nm) of HSA solutions in the presence of H_4_Bip before and after irradiation at 308 ± 10 nm, as well as a control without irradiation for the same time. As can be seen, the fluorescence intensity of the protein solution at 350 nm decreases, while the fluorescence in the region of 450 nm, corresponding to strongly fluorescent oxidized forms of H_4_Bip, increases.

The degree of oxidative modification of HSA, expressed through the fraction of oxidized protein (FOP), is presented in Table 2. As can be seen from Table 2, the greatest changes in the oxidation state of HSA (0.64 ± 0.028) occur when H_4_Bip is added to HSA, i.e., under H_4_Bip autoxidation conditions. These conditions simulate the conditions for the formation of H_2_O_2_ in vitiligo. An additional source of radicals to ROS is H_4_Bip, since radicals of a pterin nature are formed during autoxidation. They can serve as an additional source of oxidative stress for the protein. When the system (H_4_Bip + HSA) is irradiated, the degree of protein oxidation is lower (0.39 ± 0.017). The lowest degree of oxidation (0.24 ± 0.01) occurs under the direct action of UV (308 ± 10 nm) on the protein, apparently due to the excitation of protein chromophores (tryptophan and tyrosine).

#### 2.2.3. Dynamic Light Scattering Analysis of HSA Aggregates

HSA aggregation as a result of oxidative stress in the presence of H_4_Bip without irradiation and under UV irradiation (308 ± 10 nm) was studied using dynamic light scattering (DLS). Figure 9 presents the distributions of hydrodynamic radii (*R*_h_) of protein particles in the studied samples.

It should be noted that, at the studied low protein concentration (0.1 mg/mL), the distributions reflect only the aggregates present in the solution or, in the case of native HSA, oligomers (186 ± 18 nm) (Figure 9a), since small molecules of HSA itself make a low contribution to the light scattering intensity. However, we can observe the formation of larger aggregates with sizes reaching hundreds of nanometers and micrometers (≥2000 nm) upon the addition of H_4_Bip (Figure 9b,c). In the presence of an excess of H_4_Bip (Figure 9b), aggregates with *R*_h_ 514 ± 119 nm were formed, while under the action of UV in the presence of H_4_Bip (Figure 9c), a part of the protein was preserved in the form of smaller oligomers (*R*_h_ 221 ± 32 nm). This may be due to the fact that the effect of pterin radicals on HSA is reduced due to the formation of dihydropterin dimers from H_4_Bip. The existence of an additional peak with *R*_h_ 47 ± 5 nm (Figure 9b) can also be due to increased aggregation of native HSA in the presence of H_4_Bip. It is remarkable that this peak is absent after UV irradiation in the presence of H_4_Bip.

## 3. Discussion

We believe that the pathology of vitiligo is based on the formation of an autocatalytic cycle of excessive synthesis of H_2_O_2_, leading to oxidative stress and triggering of melanocyte apoptosis. Where does the excess H_2_O_2_ come from? It can occur with an excessive H_4_Bip, which in vitiligo is 3–5 times higher than normal and inevitably undergoes autoxidation in the presence of molecular oxygen. This results in the formation of radical intermediate products of a pterin nature, oxidized forms of pterins and H_2_O_2_ (see Figure 2). H_2_O_2_ triggers an autocatalytic cycle for the synthesis of excessive hydrogen peroxide through cytokines, in particular through γ-interferon. This activates the interferon-inducible enzyme guanosine triphosphate cyclohydrolase, which synthesizes an excess of H_4_Bip. Thus, strong oxidative stress (10^−3^ M H_2_O_2_) occurs in melanocytes.

Using quantum-chemical calculations, we have evaluated the feasibility of reactions participating in H_4_Bip autoxidation and photooxidation. As a whole, H_4_Bip photodegradation occurs according to both type-I and type-II mechanisms. The conclusions about the feasibility of certain “dark” and light reactions are in line with our previous studies [31,38]. Among the reactions of (H_2_Ptr)_2_ dimer formation, the reaction between qH_2_Ptr and ^3^H_2_Ptr* triplet seems to be the most feasible. The energy profiles of all the described reactions are summarized in Figure 5.

We have simulated the conditions of oxidative stress with an excess of H_4_Bip using HSA as a model of the H_4_Bip protein environment. Currently, the oxidative modification of proteins, and in particular the free radical oxidation of HSA, is being widely studied. HSA is considered as a target for free radicals and as a marker of oxidative stress [44,45,46]. Pterin radicals generated during H_4_Bip photooxidation can function as photosensitizers of protein oxidation [47,48]. The radical trap in HSA is the thiol S-H group of Cys-34 [49]. The second amino acid sensitive to the action of free radicals is methionine. HSA contains six methionine residues. The third target for free radicals in HSA is 18 tyrosines and 1 tryptophan, which fluoresce under UV irradiation [46,50,51]. Therefore, it is possible to study the oxidative degradation of amino acids in a protein using a highly sensitive UV fluorescence method.

It has been shown that the fluorescence of HSA in the presence of H_4_Bip and oxygen strongly decreases: FOP is 0.64 ± 0.028, and hence, more than half of the chromophores are subjected to destruction. This seems to be due to the fact that the protein is a target for free radicals of both a pterin nature [52] and ROS, which are generated during H_4_Bip autoxidation.

When the (H_4_Bip+ HSA) system is exposed to UV (308 ± 10 nm), dihydropterin dimers are formed in parallel with the oxidative degradation of the protein by pterin radicals, ROS, and H_2_O_2_. As a result of the formation of dimers, H_4_Bip leaves the system and a smaller amount of it is involved in the autoxidation, and FOP decreases to 0.39. This simulates the redox state during UVB therapy of vitiligo when H_4_Bip leaves the reaction zone, turning into dimers. In the case where only protein is exposed to UV irradiation in the presence of oxygen, we have the lowest FOP (0.24 ± 0.01). Apparently, this is due to the fact that the irradiation source (308 ± 10 nm) had a spectral slit width of 20 nm and had 298 nm radiation, which primarily excited tryptophan residues in the protein and they were oxidized by oxygen. The first electronic absorption band of tryptophan S_0_
→ S_1_ is in the region of 260–310 nm. Thus, using the UV fluorescence method in the (H_4_Bip + HSA) system, we were able to simulate the relative levels of oxidative stress that occur during the autoxidation and photooxidation of H_4_Bip under conditions of it being present in 3–5-fold excess in vitiligo. At the same time, it was noted that UVB radiation at 308–311 nm has the disadvantage, in that it practically captures the protein excitation region and can lead to oxidative degradation of proteins. In this regard, we have proposed a light-emitting diode with λ_max_ 325 nm as a source of radiation for phototherapy of vitiligo [28].

Oxidative modification of proteins usually leads to a change in the conformation of the molecule due to the formation of dityrosines, and the oxidation of tryptophans and cysteines. These destructions lead to the disappearance of the α-helical structure of the protein and the appearance of aggregates. The DLS method was used to study HSA aggregation in the presence of H_4_Bip. Studies of the distribution of hydrodynamic radii of protein particles showed that in the presence of H_4_Bip in the dark and under UV irradiation, HSA aggregation occurs with a significant increase in particle size, up to hundreds of nanometers and micrometers (≥2000 nm) (Figure 9b,c). In the case of HSA, in the presence of an excess of H_4_Bip, the oxidation products which are a source of radicals (Figure 9b), aggregates with *R*_h_ 514 ± 10 nm were formed, while under the action of UV in the presence of H_4_Bip, a significant part of the protein was preserved in the form of smaller oligomers (*R*_h_ 221 ± 10 nm). The smaller aggregates of apparently native HSA (*R*_h_ 47 ± 5 nm) are also formed in the presence of H_4_Bip without irradiation and do not appear under UV irradiation. This may be due to the fact that the effect of pterin radicals on HSA is reduced due to the transition of H_4_Bip into dihydropterin dimers. The HSA aggregates formed, and the aggregates with *R*_h_ ≥ 2000 nm can become immunogenic particles upon the interaction with pterin radicals and the highly reactive quinonoid form of dihydropterin. It is known that H_2_O_2_ triggers the systems of humoral and cellular immunity, which can respond to emerging immunogenic complexes and lead to the death of melanocytes.

## 4. Materials and Methods

### 4.1. Quantum Chemical Calculations: Conformational Analysis, Geometry Optimization, and Hessian Calculation

Conformational analysis of H_4_Bip and its derivatives has been performed in Spartan v. 20 (http://www.wavefun.com, accessed on 30 August 2023). The series of low-energy conformers has been generated using molecular mechanics, namely the MMFF force field [53]. Selection of the most stable conformer has been carried out based on the Hartree–Fock method and second-order Møller–Plesset perturbation theory (MP2), implemented in the ORCA program v. 3.0 [54]. Thus, the MMFF-generated geometries of H_4_Bip and its derivatives have been optimized, and the total energy has been calculated using the MP2/6-31G(d)+ method; thus, the most stable conformers have been selected. Then, further optimization and Hessian calculation have been performed using the MP2/6-31G(d,p)+ method to obtain the Gibbs free energy. The COSMO continuum model [55] has been used to take into account a water solvent with a dielectric constant ε equal to 80.

### 4.2. Oxidative Modifications of HSA in the Presence of H_4_Bip

5,6,7,8-tetrahydro-L-biopterin (H_4_Bip), H_2_Bip and other pterins have been purchased from “Schirks Laboratories” (Zurich, Switzerland). Other reagents used in the article have been obtained from “Sigma-Aldrich Co” (Saint Louis, MO, USA).

We have used freshly prepared solutions of 0.1 mg/mL (1.5 × 10^−6^ M) HSA in 0.01 M phosphate buffer (pH 7.0) in the absence or presence of 1.1 × 10^−4^ M H_4_Bip, as well as solutions of 1.6 × 10^−4^ M H_4_Bip in 0.01 M Tris buffer (pH 7.0). The concentration of H_4_Bip has been determined based on the molar extinction coefficient (ε_297_ = 10,200 M^−1^cm^−1^ for pH 7.0). Irradiation of 1.5 mL solutions has been carried out in cuvettes with an optical path length of 1 cm, in 2 min steps for 16 min, at 308 ± 10 nm (41.4 W·m^−2^) with constant stirring in air. At each step, the absorption and fluorescence spectra of the studied samples were measured. The FluoroMax 4 spectrofluorimeter (Horiba Scientific, Kyoto, Japan) has been used as an UV light source as described in our previous work [28]. Controls without irradiation have been performed in parallel to each experiment.

The absorption spectra of the irradiated samples in the range of 200–500 nm have been recorded using the Shimadzu UV-1601 spectrophotometer (Kyoto, Japan). The concentration of dihydropterin dimers has been determined from the change in the absorption of the band in the 245 nm region and has been calculated taking into account the extinction coefficient (ε_245_ = 27,000 M^−1^cm^−1^ for pH 7.0 [28]).

Fluorescence spectra have been recorded upon excitation at 280 nm and extinction at 290–550 nm using the FluoroMax 4 spectrofluorimeter (Horiba Scientific, Kyoto, Japan). In order to assess the degree of protein oxidative modification under these conditions, FOP has been calculated using the formula (I_o_ − I)/I_o_, where I_o_ is the fluorescence intensity at 350 nm (with excitation at 280 nm) before irradiation, and I is the fluorescence intensity after irradiation or in dark control for the same time.

Dynamic light scattering measurements have been performed using the Photocor Complex correlation spectrophotometer (PhotoCor Instruments, Inc., College Park, MD, USA) equipped with a 632.8 nm He-Ne laser and a temperature controller. Scattered light has been collected at an angle of 90°. All measurements have been carried out at 25 °C in capped glass vials (40 × 8.2 mm). Protein samples were added to the buffer, preliminarily thermostated in the vial at 25 °C, to the final protein concentration of 0.1 mg/mL. The sample volume was 0.35 mL. The accumulation time for the autocorrelation function was no less than 60 s. The particle size distributions have been calculated using the DynaLS v.2 software. The refraction index of buffer at 25 °C is considered as equal to 1.3324, and the dynamic viscosity is 0.8912 mPa·s.

The experimental results are the arithmetic mean of at least three independent replicates with a 95% confidence interval. The results of the experiment are expressed as mean ± standard deviation.

## 5. Conclusions

We have estimated the feasibility of the reactions of H_4_Bip autoxidation and photooxidation in the presence of molecular oxygen from a theoretical viewpoint. We have also considered the reaction of dihydropterin dimer formation upon UV irradiation. The autoxidation occurs according to the experimentally established scheme. Considering the photooxidation of H_4_Bip, we have shown that photooxidation by Ptr is theoretically probable. Table 1 shows the main reactions participating in the photooxidation process: reactions of the radical chain process initiation, prolongation, and termination. Finally, we have evaluated the possibility of H_2_Ptr dimerization reactions. Apparently, the quinonoid form of H_2_Ptr and the triplet state of benzoid form of H_2_Ptr play a significant role in this process. The participation of an intermediate complex of the quinhydrone type, which is formed between the quinonoid and benzoic forms of dihydropterin in the reaction of dimer formation that facilitates this process, is discussed.

Theoretical calculations are consistent with the obtained experimental data, which have shown the formation of dihydropterin dimers under the action of light, regardless of the presence of the protein in the system. Also, the presence of radicals of a pterin nature and ROS during the oxidation of H_4_Bip is confirmed by the fact that oxidative modification of HSA occurs with further aggregation of the oxidized protein, which has been shown using fluorescence methods with FOP calculation and DLS. Under the action of UV irradiation in the presence of H_4_Bip, the degree of oxidative modification of the protein has been lower, which may be due to the phototransformation of H_4_Bip into dihydropterin dimers. It can be assumed that similar processes can occur in a cell during UV phototherapy of vitiligo. The results can be presented in the form of a figure (Figure 10).

## Figures and Tables

**Figure 1 ijms-24-13586-f001:**
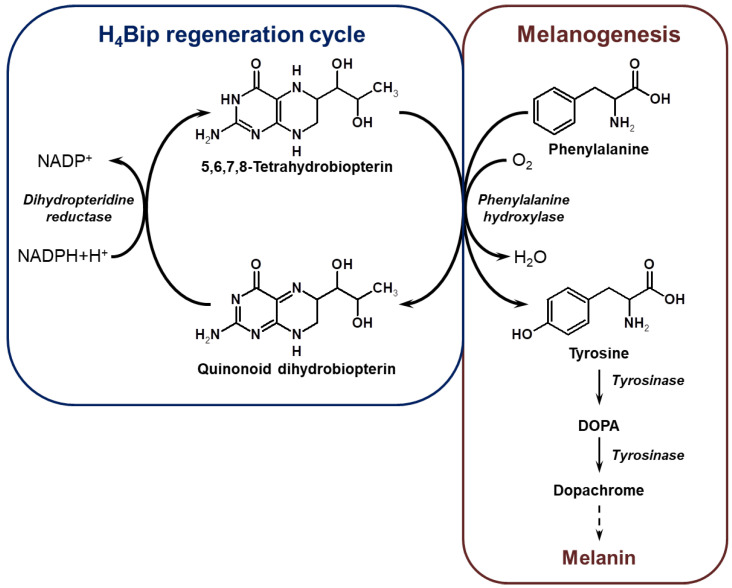
Melanin biosynthesis and the H_4_Bip regeneration cycle.

**Figure 2 ijms-24-13586-f002:**
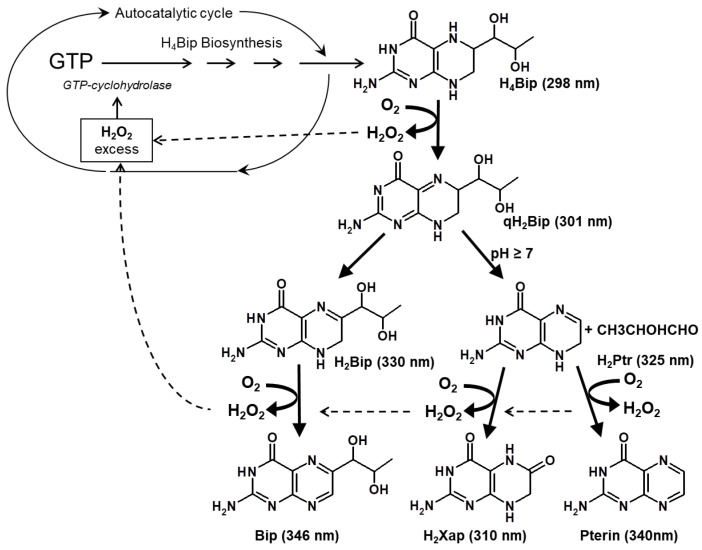
Scheme of tetrahydrobiopterin autoxidation and autocatalytic cycle closure in vitiligo (GTP—guanosine triphosphate; qH_2_Bip—quinonoid 6,7-dihydrobiopterin; H_2_Bip—7,8-dihydrobiopterin; H_2_Ptr—7,8-dihydropterin; Bip—biopterin; H_2_Xap—dihydroxanthopterin). The long-wavelength maxima of the absorption spectra are indicated in brackets [28].

**Figure 3 ijms-24-13586-f003:**
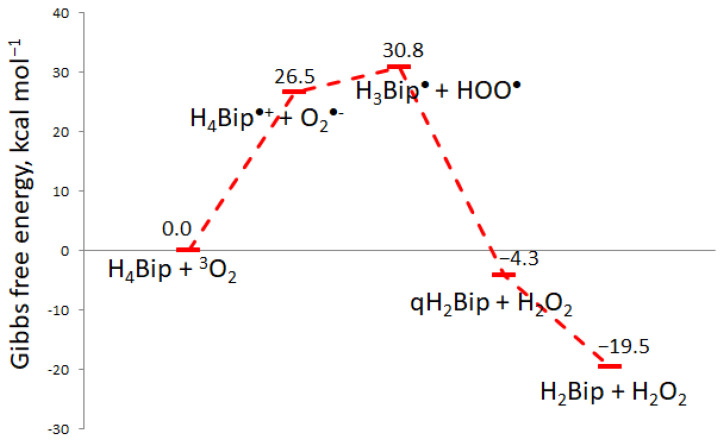
The Gibbs free energy (∆_f_Gº) profile of the reaction between H_4_Bip and molecular oxygen.

**Figure 4 ijms-24-13586-f004:**
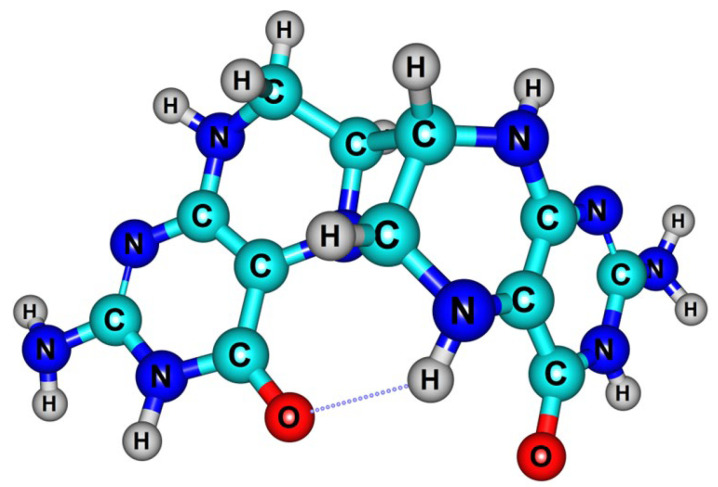
Optimized geometry of the (H_2_Ptr)_2_ azacyclobutame dimer (the MP2/6-31G(d,p)+ method, COSMO water).

**Figure 5 ijms-24-13586-f005:**
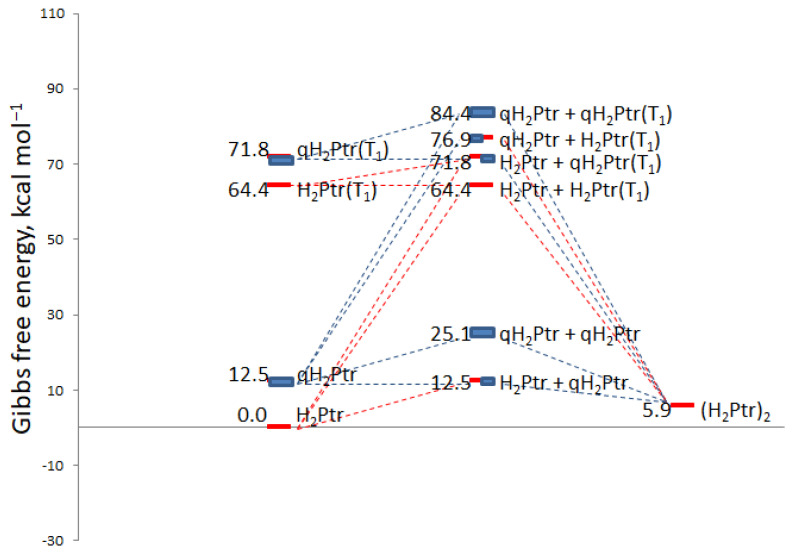
Energy profiles of the H_2_Ptr dimerization reactions. Photoreactions are considered along with “dark” reactions.

**Figure 6 ijms-24-13586-f006:**
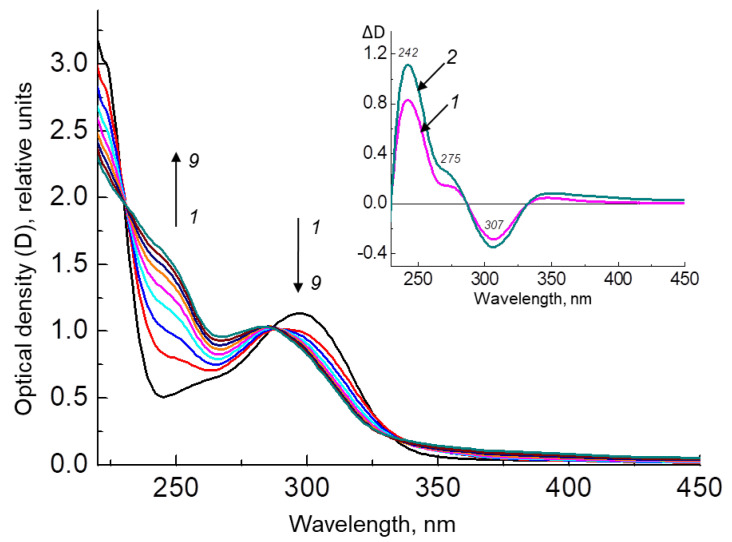
Changes in the absorption spectra of a solution of 1.5 × 10^−6^ M HSA in 0.01 M phosphate buffer (pH 7.0) in the presence of 1.1 × 10^−4^ M H_4_Bip under stepwise irradiation at 308 ± 10 nm in the presence of atmospheric oxygen. Irradiation time: 1—0 min; 2—2 min; 3—4 min; 4—6 min; 5—8 min; 6—10 min; 7—12 min, 8—14 min, 9—16 min. The inset shows the difference spectrum of light minus the initial one, exposure time: 1—8 min, 2—16 min.

**Figure 7 ijms-24-13586-f007:**
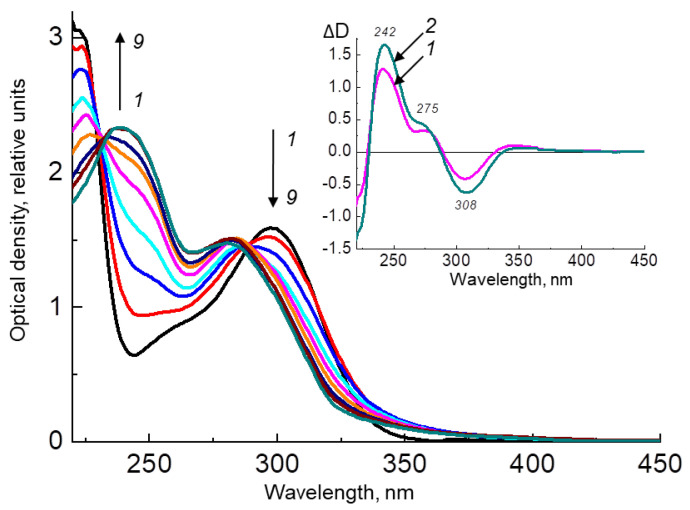
Changes in the absorption spectra of a solution of 1.6 × 10^−4^ M H_4_Bip in 0.01 M Tris buffer (pH 7.0) under stepwise irradiation at 308 ± 10 nm in the presence of atmospheric oxygen. Irradiation time: 1—0 min; 2—2 min; 3—4 min; 4—6 min; 5—8 min; 6—10 min; 7—12 min, 8—14 min, 9—16 min. The inset shows the difference spectrum of light minus the initial one, exposure time: 1—8 min, 2—16 min.

**Figure 8 ijms-24-13586-f008:**
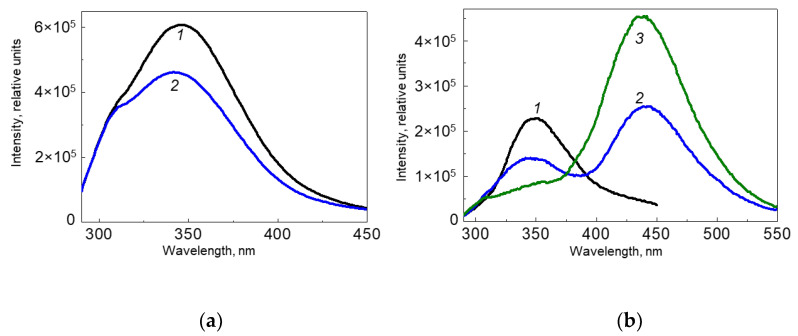
Changes in the fluorescence spectra of a solution of HSA in 0.01 M phosphate buffer pH 7 (**a**) in the absence of H_4_Bip and (**b**) in the presence of H_4_Bip and atmospheric oxygen upon excitation at 280 nm: 1—initial solution (0 min); 2—after irradiation at a wavelength of 308 ± 10 nm for 16 min; 3—control without irradiation (16 min).

**Figure 9 ijms-24-13586-f009:**
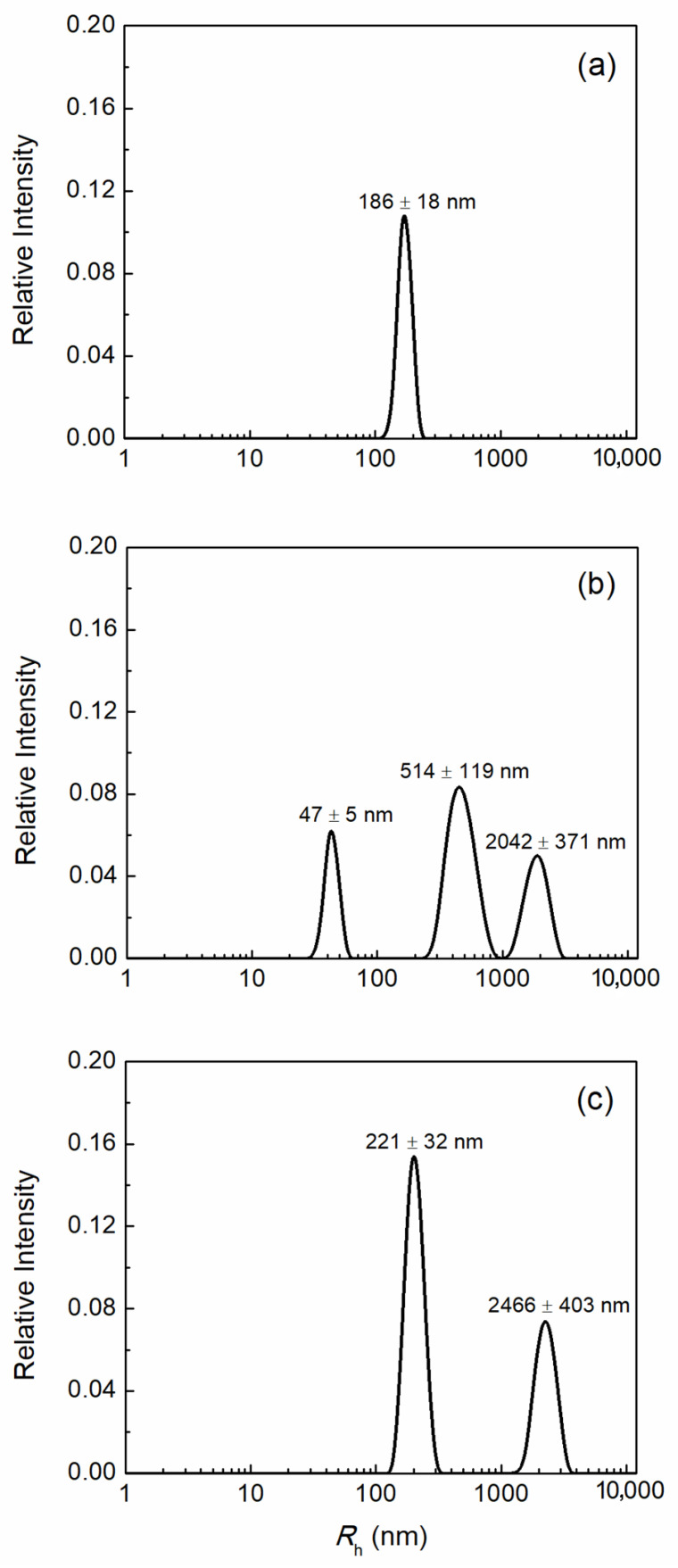
Distribution of hydrodynamic radii (*R*_h_) for solutions of 0.1 mg/mL HSA in 0.01 M phosphate buffer, pH 7.0 (**a**) HSA; (**b**) HSA in the presence of H_4_Bip without irradiation; (**c**) HSA in the presence of H_4_Bip after irradiation. Irradiation was carried out at 308 ± 10 nm for 16 min.

**Figure 10 ijms-24-13586-f010:**
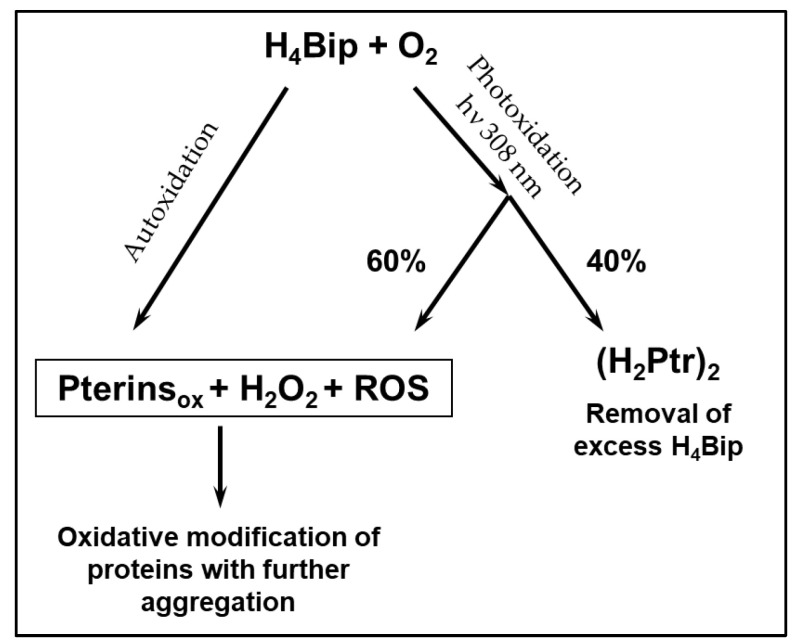
Pathways of H_4_Bip transformation during autoxidation and photooxidation (in vitiligo phototherapy).

**Table 1 ijms-24-13586-t001:** Feasibility of H_4_Bip photooxidation reactions. The Gibbs free energy of the reactions is calculated with the MP2/6-31G(d,p)+ method. COSMO water with a dielectric constant equal to 80 is the solvent.

Reaction			∆_f_Gº, kcal mol^−1^
^3^Ptr* + H_4_Bip → Ptr^•−^ + H_4_Bip^•+^	(3)	chain initiation	−21.4
Ptr^•−^ + O_2_ → Ptr + O_2_^•−^	(16)	chain initiation	−30.4
H_4_Bip + HOO^•^ → H_3_Bip^•^ + H_2_O_2_	(7)	chain prolongation	−18.0
H_3_Bip^•^ + O_2_ → H_2_Bip + HOO^•^	(8)	chain prolongation	−1.5
2H_3_Bip^•^ → H_2_Bip + H_4_Bip	(12)	chain termination	−32.4
2HOO^•^ → H_2_O_2_ + O_2_	(13)	chain termination	−48.9

**Table 2 ijms-24-13586-t002:** Calculation of the fraction of oxidized protein (FOP) for HSA under irradiation at 308 nm in the presence and absence of H_4_Bip.

Sample	FOP	Irradiation Time, min
HSA	0.24 ± 0.010	16
H_4_Bip + HSA	0.39 ± 0.017	16
H_4_Bip + HSA	0.64 ± 0.028	0 *

*—control without irradiation, 16 min

## Data Availability

Data are contained within the article.

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
