# Peer review of "Tetrahydrobiopterin as a Trigger for Vitiligo: Phototransformation during UV Irradiation"

_ijms, 2023, doi:10.3390/ijms241713586_

Round 1

Reviewer 1 Report

This paper by Telegina et al investigates the reaction of tetrahydrobiopterin (H4Bip) in the dark and under UV irradiation by a combined computational and experimental approach. The aim of the study is to assess the role of H4Bip in triggering vitiligo starting from the increased levels of this cofactor in vitiligo melanocytes.  The theory of the central role of H4Bip in vitiligo has been put forward several years ago by Schallreuter and coworkers and has been  reconsidered and reinvestigated from time to time. The autooxidation of H4Bip would give rise of increased level of H2O2 and ROS  leading to the onset of a chronic oxidative stress conditions.

The aus have already published several works on this topic and the present one appears as a further contribution where they systematically examine the feasibility under a thermodynamical point of view of the reaction of H4Bip with oxygen and the processes occurring upon photoirradiation.  The presentation of the possible reactions for which Gibbs energy is calculated is sufficiently clear and summarized in schemes. Also, the experimental part of the study is properly designed and presented. In general, the starting hypothesis and results obtained are reasonable and sound

I have only a few remarks

In the first part of the study that is computational analysis of auto/photoirradiation reactions the kinetics  issues are never considered or discussed though this is obviously very critical parameter to assess the actual impact of a given process e.g. under the condition of PUVA phototherapy.

Paragraph 2.2.1 the aus refer to an intermediate intermolecular complex and assign an absorption max to this species (307 nm). Please explain on which basis this was done

An overall final scheme summarizing the processes investigated including the oxidation of BSA in the presence of H4Bip w/wo irradiation would contribute to the clarity of the presentation

Author Response

Please, find step-by-step answers in a file attached to this message.

Reviewer 2 Report

The manuscript by Telegina et al. entitled “Tetrahydrobiopterin as a trigger for vitiligo: phototransformation during UV-irradiation’ concerns theoretical and experimental investigations of the role of tetrahydrobiopterin on occurrence of vitiligo. This manuscript is a continuation of previous works of the Authors in that matter. In the current paper Authors proposed the molecular reactions which may be present when vitiligo is treated with UV light.

Please find below my comment:

1. Materials and methods are insufficiently described.  In point 4.2 please provide proper experimental protocols for the performed studies, if known methods were used please provide appropriate references. Some information is missed eg.: volume of samples, number of replicates. Please provide the methods in the way, that the experiments could be repeated in other laboratory.

2. Did Authors implemented any statistical analysis? Please provide details.

3. In introduction please add some information of other Author.

4. Please correct line 63, page 3: units after 280-320

Author Response

Please, find the responses attached.

Round 2

Reviewer 2 Report

Authors implemented all recommended changes. The manuscript could be published in current form.